# Lipid-Based Nanoparticles Fused with Natural Killer Cell Plasma Membrane Proteins for Triple-Negative Breast Cancer Therapy

**DOI:** 10.3390/pharmaceutics16091142

**Published:** 2024-08-29

**Authors:** Eun-Jeong Won, Myungchul Lee, Eui-Kyung Lee, Seung-Hoon Baek, Tae-Jong Yoon

**Affiliations:** 1Research Institute of Pharmaceutical Science and Technology (RIPST), Department of Pharmacy, Ajou University, 206 Worldcup-ro, Yeongtong-gu, Suwon 16499, Republic of Korea; ejwon@kribb.re.kr; 2Nucleic Acid Therapeutics Research Center, Korea Research Institute of Bioscience and Biotechnology (KRIBB), 30 Yeongudanji-ro, Ochang, Cheongwon, Cheongju 28116, Republic of Korea; 3School of Pharmacy, Sungkyunkwan University, 2066 Seobu-ro, Jangan-gu, Suwon 31065, Republic of Korea; leemc33@daum.net (M.L.); ekyung@skku.edu (E.-K.L.); 4Department of BioHealth Regulatory Science, Graduate School of Ajou University, 206 Worldcup-ro, Yeongtong-gu, Suwon 16499, Republic of Korea; 5Moogene Medi Institute, 25, Misagangbyeonjungang-ro 7beonan-gil, Hanam 12939, Republic of Korea

**Keywords:** lipid-based nanoparticles, natural killer cells, gene delivery, ceramide, immunotherapy

## Abstract

Immunotherapy combined with chemicals and genetic engineering tools is emerging as a promising strategy to treat triple-negative breast cancer (TNBC), which is more aggressive with poorer progress than other breast cancer subtypes. In this study, lipid-based nanoparticles (LNPs) possessed an NK cell-like function that could deliver tumor-specific therapeutics and inhibit tumor growth. LNPs fused with an NK cell membrane protein system (NK-LNP) have three main features: (i) hydrophilic plasmid DNA can inhibit TNBC metastasis when encapsulated within LNPs and delivered to cells; (ii) the lipid composition of LNPs, including C18 ceramide, exhibits anticancer effects; (iii) NK cell membrane proteins are immobilized on the LNP surface, enabling targeted delivery to TNBC cells. These particles facilitate the targeted delivery of HIC1 plasmid DNA and the modulation of immune cell functions. Delivered therapeutic genes can inhibit metastasis of TNBC and then induce apoptotic cell death while targeting macrophages to promote cytokine release. The anticancer effect is expected to be applied in treating various difficult-to-treat cancers with LNP fused with NK cell plasma membrane proteins, which can simultaneously deliver therapeutic chemicals and genes.

## 1. Introduction

Triple-negative breast cancer (TNBC) is a type of breast cancer that lacks expression of estrogen receptor (ER), progesterone receptor (PR), and unamplified human epidermal growth factor receptor 2 (HER2) on the cell surface [1,2]. TNBC has a highly invasive nature. Approximately 46% of patients are diagnosed with distant metastases, and the mortality rate is 4~20% for metastatic patients in Western countries [3,4]. Due to this lack of endocrine or therapeutic molecular targets (ER, PR, and HER2), TNBC patients have limited access to adjuvant systemic chemotherapy. In addition, even a recent promising immunotherapeutic treatment using cell death ligand 1 (PD-L1) is a limited treatment method because it is applied only to 20~40% of TNBC patients with overexpression of the ligand [5,6]. Therefore, research is needed regarding new approaches to treat these difficult-to-treat TNBCs and effectively prevent metastasis.

Hypermethylated in cancer 1 (*HIC1*) is a tumor suppressor gene. Using this, several researchers have shown that epigenetic silencing can be induced through hypermethylation of the promoter region of *HIC1* in various solid tumors such as colon, liver, lung, and prostate cancers [7,8]. Loss of HIC1 can promote tumorigenesis with upregulation of stress regulatory protein SIRT1. The mechanism can evade apoptotic cell death, regulate cell cycle markers (cyclin D1 and P21), and activate tumor metastasis factors (LCN2 and VEGF) [9,10,11]. In particular, HIC1 protein expression is more significantly reduced in TNBC than in other breast cancer or normal breast cells. Researchers have proposed that restoring HIC1 expression can minimize cell invasion and metastasis in TNBC [12].

Ceramides are a family of sphingolipids consisting of sphingosine and fatty acids of various chain lengths with an endogenous form most commonly consisting of carbon chain lengths ranging from C16 to C24 in mammalian cells. C16–C24 ceramides can act as active signaling molecules that regulate various cellular processes [13]. Long-chain ceramides are the most studied sphingolipids in cancer because they are involved in apoptosis [14]. These ceramides are known to alter mitochondrial ultrastructure and reduce function and membrane potential, ultimately inducing apoptosis. Among them, C18-ceramide is catalyzed by ceramide synthases 1 and 4 in a de novo synthetic pathway. It contributes to apoptosis and inhibition of cell proliferation in HNSCC (head and neck squamous cell carcinoma) and glioma [15,16,17]. However, because it has low water solubility, it is difficult to process into cells. As a way to overcome this, LNP systems are suitable for delivering hydrophobic ceramides to cells. Therefore, adding exogenous C18-ceramide, a component of LNPs, to the lipid layer not only maintains the structure of the nanoparticles but also exerts anticancer effects.

Natural killer (NK) cells play an essential role in innate immunity, the first defense against infection and tumors through immune surveillance [18]. For example, recent clinical trials of NK cell therapy against engineered allogeneic NK cells and tumor-targeted chimeric antigen receptor (CAR)-NK cells are ongoing [19,20]. Furthermore, it has been reported that nanoparticles similar to CAR-NK cell structures can be used as artificial structures for cancer treatment strategies [21,22,23]. Nanoparticles with NK cell membrane proteins, such as tumor activation receptors (NKG2D, NKp30, NKp46), introduced onto their surface can selectively recognize most tumors. In addition to these abilities, some NK membrane receptors (RAB-10, IRGM1, RANKL Galectin-12, CB1) are known to interact with immune cells, which can be activated to induce polarization of immune cells [24]. As such, research on various anticancer applications using NK cell membrane proteins is actively underway.

In this study, we synthesized artificial nanomaterials using characteristics of NK cells that could target TNBC and investigated the possibility of targeted treatment for tumors (Figure 1). We loaded plasmid DNA-encoding HIC1 into the LNP and fused NK cell membrane proteins onto the LNP surface. Additionally, the LNPs possessed contain C18-ceramide lipids, synergistically enhancing the antitumor effect. In immunotherapy, we confirmed the ability of NK-LNPs to induce antitumor immune responses by inducing the polarization of effector cells by NK cell membrane proteins. It is expected that the synthesized nanomaterial can be developed as an artificial immunotherapy agent for solid tumors capable of complex targeted treatment.

## 2. Experimental Methods

### 2.1. Plasmid

The pET30a-recombinant human protamine (PA) plasmid was generated based on PA amino acid sequences from NCBI (#NP_002752.1). Codons were optimized for expression in *E. coli* cells and gene synthesized from Abclon. pCMV-HIC1 was purchased from Sino Biological (#HG18546-UT).

### 2.2. Expression and Purification of Recombinant Human Protamine Protein (PA)

For PA protein expression from *E. coli*, BL21 Star^TM^ (DE3) (Thermo Fisher Scientific Korea, Seoul, Republic of Korea) was transformed with pET30a-PA. Transfected cells were cultured in terrific broth until the optical density at 600 nm was 0.4–0.5. After induction with 1 mM isopropyl β-d-1-thiogalactopyranoside (IPTG), cells were incubated at 25 °C for 18 h. For PA protein purification, cultured cells were lysed with cell lysis buffer (100 mM Tris-HCl, 1 M NaCl, 20 mM imidazole, and 0.5 mM PMSF, pH 7.5). The dissolved cell solution was probe-sonicated and centrifuged at 15,000× *g* for 20 min at 4 °C. The supernatant was reacted with HisPur™ Cobalt Resin (Thermo Fisher) and washed with a column wash buffer (100 mM Tris-HCl, 0.5 M NaCl, and 40–60 mM imidazole, pH 7.5) before PA protein elution (100 mM Tris-HCl, 0.5 M NaCl, and 300 mM imidazole, pH 7.5). Eluted protein was further purified on a Strong Cation Exchange Column (Thermo Fisher), dialyzed against a storage buffer (100 mM Tris-HCl, 0.2 M NaCl, 0.1 mM EDTA, 0.5 mM PMSF, and 20% glycerol, pH 7.5), and then stored at −80 °C.

### 2.3. NK-92 Culture and Plasma Membrane Protein Isolation

The human NK cell line (NK-92) was procured from the American-Type Culture Collection (ATCC). According to the manufacturer’s culture instructions, NK-92 cells were cultured with alpha MEM (without ribonucleosides and deoxyribonucleosides, with 2 mM L-glutamine and 1.5 g/mL sodium bicarbonate), with added supplements (12.5% horse serum, 12.5% fetal bovine serum, 1% penicillin/streptomycin, 0.2 mM Myo-inositol, 0.1 mM 2-mercaptoethanol, 0.02 mM folic acid, and 100 U/mL human recombinant IL-2).

For isolation of NK cell plasma membrane proteins, cultured cells were harvested by centrifugation (125× *g*, 10 min, 4 °C). NK cell plasma membrane proteins were extracted with the Mem-PER™ plus membrane protein extraction kit (Thermo Scientific) according to the manufacturer’s instructions and then stored at −80 °C.

### 2.4. Cell Culture

All cell lines were procured from ATCC. MDA-MB-231, MDA-MB-468, and HCC38 cells were cultured in ATCC-modified RPMI medium (Gibco, Thermo Fisher Scientific Korea) and supplemented with 10% fetal bovine serum (FBS) and 1% penicillin/streptomycin (P/S). MCF10A and MCF12A were cultured in DMEM/F12 medium (Gibco) and supplemented with 5% horse serum, 20 ng/mL human recombinant EGF protein, 0.01 mg/mL insulin, 500 ng/mL hydrocortisone, and 1% P/S. MCF7 cells were cultured in DMEM medium and supplemented with 10% FBS and 1% P/S, and all cells tested negative for mycoplasma.

### 2.5. Synthesis and Characterization of NK-LNP with pDNA

Lipid nanoparticles (LNPs) were prepared using egg PC:cholesterol:C18-ceramide: DGS-NTA(Ni) (molar ratio of 1:0.2:1:0.05), dissolved in 1 mL chloroform using a thin-film hydration method. The evaporated lipid film was hydrated with 30 μM purified PA protein and 30.82 nM pCMV-HIC1 plasmid complex in 1 mL phosphate-buffered saline (PBS, pH 7.2). The LNP was formed with a hydrated lipid/PA/plasmid DNA (pDNA) mixture through a liquid nitrogen (LN_2_) freezing–thawing procedure. For NK-LNP formation, the assembled LNP was further frozen and thawed in the presence of isolated NK cell membrane proteins (total lipid: NK cell membrane protein = 1:5, *w/w*). Excess PA/pDNA/NK cell membrane proteins were removed with a CL-4B Sepharose size extraction column. The resultant NK-LNP with pDNA solution was filtered through a 0.2 μm syringe filter and then kept at 4 °C. NK-LNP with pDNA morphology was observed using cryo-transmission electron microscopy (cryo-EM, Tecnai F20 G2, FEI). The hydrodynamic size and zeta potential of the NK-LNP with pDNA were characterized by a Zetasizer (Nano ZS, Malvern Instrument, Malvern, UK). Particle concentration was measured by nanoparticle tracking analysis (NTA, Nanosight NS3000, Malvern Instruments). The fused NK cell membrane proteins and encapsulated PA/pDNA complexes of the NK-LNP were qualitatively analyzed by Western blotting after being powdered with a freeze-dryer and lysed in a membrane protein extraction buffer. Encapsulated pDNA was quantified by electrophoresis before removing the non-conjugated pDNA with PA.

### 2.6. Targeting and Transfection Efficiency of NK-LNP

MCF12A, MDA-MB-231, and MDA-MB-468 cells were seeded into 6-well cell culture plates (0.35 × 10^6^ cells) at 37 °C under 5% CO_2_. Depending on experimental conditions, cells were incubated with the NK-LNP (RITC-labelled PA/pDNA) (6.57 × 10^10^ ± 3.12 × 10^9^ particles in 1 mL opti-MEM). First, to determine the TNBC targeting efficacy of the NKG2D receptor, cells were incubated with an NKG2D ligand blocker (anti-ULBP2/5/6, R&D Systems). After fixation, cells were treated with particles at 37 °C for 1 h and analyzed by flow cytometry (Accuri C6 Plus, BD, Franklin Lakes, NJ, USA). To prove receptor-mediated endocytosis of NK-LNP(PA/pDNA), TNBC was added to genistein (10 μg/mL), chlorpromazine (10 μg/mL), nocodazole (1.5 μg/mL), and cytochalasin B (2.4 μg/mL). NK-LNP(PA/pDNA) treated cells were then incubated at 37 °C for 24 h (except for samples at 4 °C) and analyzed by flow cytometry.

### 2.7. Fluorescence Image Analysis

To visualize endosomal escape and nuclear delivery, MDA-MB-468 cells were seeded into 8-well glass chamber slides (0.025 × 10^6^ cells) at 37 °C under 5% CO_2_. The cells were treated with NK-LNP(RITC-labeled PA/pDNA) at the same concentration as in the previous experiment. Endosome escape was also performed according to the manufacturer’s instructions (Invitrogen Korea, Seoul, Republic of Korea). To confirm nuclear localization, the NK-LNP(RITC-labeled PA/pDNA)-treated cells were fixed with 4% paraformaldehyde, permeabilized with 0.01% Triton X-100, and blocked with 3% BSA in TBST. The cells were then incubated at 4 °C with an anti-CD56 antibody (Cell Signaling, Danvers, MA, USA) followed by fluorescence-conjugated secondary antibodies. The cells were observed using a cell automated microscope (Lionheart FX, BioTek, Winooski, VT, USA).

### 2.8. Western Blot Analysis

MDA-MB-468 cells were seeded in 60-mm^2^ cell culture plates (1.3 × 10^6^ cells) at 37 °C under 5% CO_2_, followed by incubation with NK-LNP(PA/pDNA) (2.44 × 10^11^ ± 1.19 × 10^10^ particles in 2.5 mL opti-MEM) for 6 h at 37 °C. These cells were lysed in RIPA buffer with a protease and phosphatase inhibitor (Sigma Aldrich Korea, Seoul, Republic of Korea). Whole-cell lysates were quantified by a BCA protein assay, loaded onto 12% SDS-PAGE gels under denaturing conditions, and transferred onto PVDF membranes (iBlot2 Dry Blotting System, Thermo Fisher Scientific). Membranes were blocked with 5% skim milk and incubated overnight at 4 °C with the following primary antibodies: anti-Na-K ATPase, p21CIP, Cyclin D1 (Abcam, Dawinbio Korea, Hanam, Gyeonggi-do, Republic of Korea), anti-HIC1, NKG2D (Anova, Taipei City, Taiwan), anti-CD56, phosphor-NFκB, cleaved caspase3 (Cell Signaling, Danvers, MA, USA), anti-CD45, NP62, SIRT1, PCNA, BCL2, BAX, and actin (Santa Cruz, Koram Biotech, Seoul, Republic of Korea). Membranes were incubated with HRP-conjugated secondary antibodies and developed with Western ECL substrate. Luminescent images were analyzed using an LAS500 (GE Healthcare Korea, Seoul, Republic of Korea).

### 2.9. Cell Viability, Migration, Invasion, and Apoptosis Assay

MDA-MB-468 cells and MCF12A were seeded in 96-well plates (1.5 × 10^4^ cells/well) and treated with NK-LNP(PA/pDNA) synthesized under various conditions. The cells were incubated with WST-8 reagent (Dojindo) for 1 h, and the absorbance was measured using a microplate reader (BioTek, Winooski, VT, USA) at 450 nm every 24 h. To estimate cell migration and invasion capability of NK-LNP(PA/pDNA), MDA-MB-468 cells were seeded in 12-well plates (0.15 × 10^6^ cells/well), treated with NK-LNP(PA/pDNA) without C18-ceramide, and analyzed following the manufacturer’s instructions (Komabiotech, Seoul, Republic of Korea). To evaluate cellular apoptosis depending on NK-LNP(PA/pDNA) with or without C18-ceramide, MDA-MB-468 cells were treated under various experimental conditions for 48 h. They were then stained with an EzWay Annexin V-FITC Apoptosis Detection Kit (Komabiotech).

### 2.10. THP1 Immune Cell Real-Time PCR Analysis and Cytokine Detection

THP-1 cells (human monocytes) were obtained from the Korean Cell Line Bank (KCLB). These THP-1 cells were seeded (1.3 × 10^6^ cells) and differentiated into M0-macrophages by phorbol 12-myristate 13-acetate (PMA, 65 ng/mL) at 37 °C for 24 h [25,26]. After that, M0-macrophages were treated with NK-LNP(PA/pDNA) for 48 h (Appendix A). THP-1 cells were also seeded (1.3 × 10^6^ cells) and differentiated into immature dendritic cells in the presence of human recombinant IL-4 protein (0.65 μg/mL) and GM-CSF (0.65 μg/mL) at 37 °C for 5 days [27]. After that, immature dendritic cells were co-cultured with NK-LNP(PA/pDNA) and treated with TNBC (Appendix A). After 48 h, THP-1 cells were harvested, and total RNA was extracted using TRIzol (Invitrogen). Real-time PCR was carried out in Quantstudio3 (Applied Biosystems, Thermo Fisher Scientific Korea) using SYBR Green premix (Thermo). Real-time PCR primer sequences are listed in Appendix A. Secreted cytokines were quantified using ELISA (human TNFα, IL-6, IL-10, and IL-12 ELISA kit; Abfrontier, Seoul, Republic of Korea) according to the manufacturer’s assay procedure.

### 2.11. Statistical Analysis

All experiment data are expressed as mean ± standard error of the mean (SEM). Statistical results were analyzed using an unpaired two-tailed Student’s *t*-test. All analyses were conducted using GraphPad Prism 8. Statistical details are described in figure legends.

## 3. Results

### 3.1. Isolation of NK Cell Plasma Membrane Proteins and Complexes with Protamine

In tumor cells, ligands of ULBP1, ULBP2/5/6, ULBP3, ULBP4, and MICA/B can bind to NKG2D receptors among tumor-activating receptors of NK cells [28]. We confirmed that TNBC (MDA-MB231 and MDA-MB-468) had a higher expression level of NKG2D ligands than normal breast cells (MCF12A) (Appendix A). This suggests that TNBC cells are susceptible to NKG2D receptors. Plasma proteins were separated to extract membrane proteins from activated NK92 cells. Various membrane proteins, such as CD56, NKG2D, NCR1 (NKp46), NCR2 (NKp44) and NCR3 (NKp30), were confirmed through dot–blot analysis (Appendix A).

Polyarginine refers to peptides or proteins with various cationic properties and can bind to anionic plasmid DNA through cationic charge–anionic charge interactions. Therefore, polyarginine collaborates to introduce plasmids into LNPs with anionic overall charges. For this purpose, we produced cationic protamine (PA). PA was tagged at the N and C termini using a sequence obtained only from human sperm. The histidine tag facilitates protein purification and allows binding to DGS-NTA(Ni), a component of LNP. Cationic protamine (PA), with excellent biocompatibility, has been used as an effective method to encapsulate anionic plasmid DNA (pDNA) into LNPs [29,30]. We further modified two functional peptide sequences to increase aqueous solution stability during hybridization with pDNA and LNP loading (Appendix A). The recombinant human PA with 15 kDa was codon-optimized and purified from *Escherichia coli* (*E. coli*) (Appendix A). Electrophoretic mobility shift assay (EMSA) confirmed electrostatic interactions between modified recombinant PA and pDNA (Appendix A). DGS-NTA-Ni is well known to form a chelating bond with histidine [31]. In brief, we designed PA/pDNA, which is bound by electrostatic attraction, to bind to DGS-NTA-Ni, a component of LNP, through the histidine tag at the N- and C-termini of PA. Therefore, PA and DNA are bound by electrostatic attraction to form a hybrid, and PA/DNA is loaded onto DGS-NTA-Ni, a component of LNP, through the His tag of PA. We also added an S-tag to the PA sequence to promote the expression of polyarginine residues in *E. coli* and improve PA/pDNA complex stability compared to un-modified protamine sulfate (PS) proteins (Appendix A).

### 3.2. Synthesis and Characterization of NK-LNP(PA/pDNA)

NK-LNP(PA/pDNA) was synthesized with the following three steps: PA/pDNA complexation, encapsulation of PA/pDNA into LNP, and surface fabrication with NK cell membrane proteins onto the LNP (Figure 1A). First, the PA/pDNA complex was hydrated with a thin film of LNP components of egg PC, cholesterol, DGS-NTA-Ni, and C18-ceramide, similar to previous methods [31]. Recombinant human PA/pDNA-encapsulated LNPs showed superior solution stability to LNPs containing salmon-derived protamine sulfate (PS). They also had a uniform size and poly-dispersity index (Appendix A). Assembled LNPs were then further processed in the presence of isolated NK cell membrane proteins. NK-LNP(PA/pDNA) formed protein–lipid interactions through hydrophobic interaction. These are presumed to be hydrophobic portions of integral membrane proteins embedded within phospholipids, such as in the fluid mosaic model. The shape, size, surface charge, and particle concentration of NK-LNP(PA/pDNA) were determined after extraction using a CL4B size extraction column and 0.2 µm filtration. NK cell membrane protein fusion-LNP(PA/pDNA) showed a spherical morphology in cryo-EM analysis. Compared to LNP(PA/pDNA), this appearance was expressed as a slightly darker contrast on the surface due to fused proteins (Figure 1B). The size of NK-LNP(PA/pDNA) was 130 ± 7.6 nm (lipid: NK cell membrane protein = 1:5, *w*/*w*), which was not significantly affected by NK cell membrane protein modification (Figure 1C). The zeta potential value decreased to −17 mV depending on the proportion of membrane proteins used for surface modification (Figure 1D). Compared with the zeta potential of NK cell membrane protein alone of −22 mV, NK-LNP(PA/pDNA) values were slightly different. The protein profile of NK-LNP(PA/pDNA) was confirmed by SDS-PAGE and Western blot analysis. Isolated membrane proteins were mainly retained on the surface of NK-LNP(PA/pDNA), indicating that LNPs were fully fused at a lipid-to-protein reaction ratio (1:5, lipid: protein, *mg:mg*) (Figure 1E). In Figure 1F, it was confirmed that the binding of the loaded pDNA/PA complex was maintained stably even when NK membrane proteins were fused to the surface of the LNP. After purification, membrane proteins from CD56, Na-K pump, CD45, and NKG2D receptor were detected in the NK-LNP particle fusion. Additionally, their concentration was confirmed to have increased gradually at a reaction ratio of 1:1, 1:5, or 1:10 (lipid: protein, *w*:*w*). On the other hand, signals from nuclear protein (NP62) and cytoplasmic protein (Actin) were not detected in the purified NK-LNP(PA/pDNA). This confirmed that excess NK cell membrane proteins not bound to LNPs were well separated after purification. The pDNA encapsulation efficiency (E.E. %) was quantified by measuring the amount of remaining pDNA not bound to NK-LNP(PA/pDNA) by agarose gel electrophoresis. The E.E. % was calculated using the formula: (E.E. %) = (W_I_ − W_R_)/W_I_ × 100, where W_R_ was the amount of unbound pDNA in particles and W_I_ was the total quantity of initial adding pDNA. The pDNA encapsulation efficiency increased proportionally to the DGS-NTA-Ni concentration of NK-LNP(PA/pDNA). It was verified that 73% of pDNA was encapsulated (Appendix A).

### 3.3. Specific Targeting Ability and Cellular Uptake Mechanism

First, we determined whether the NKG2D receptor on NK-LNPs could target TNBC cells. MCF12A, MDA-MD-231, and MDA-MD-468 cells were incubated with NK-LNP (RITC-labelled PA/pDNA) at 37 °C for 1 h (Figure 2A). The targeting efficacy of NK-LNPs in the MDA-MB-468 was quantitatively evaluated at 72% through FACS analysis. As a control, the targeting efficacy of MDA-MB-468 cells pretreated with the NKG2D ligand blocker (anti-ULBP2/5/6) was reduced to less than 14%. MDA-MB-231 cells also showed the same pattern of reduction. This confirmed that TNBC targeting was possible through the NKG2D ligand of cells with the NKG2D receptor protein of NK-LNP. Additionally, it was confirmed again that targeting was not performed in MCF12A cells without NKG2D ligand as a negative cell line, regardless of using a blocker. We performed cellular uptake mechanism analysis at 37 °C for 24 h to demonstrate receptor-mediated endocytosis of NK-LNP(PA/pDNA) (Figure 2B). A clathrin-mediated endocytosis inhibitor (chlorpromazine) and 4 °C incubation reduced NK-LNP(PA/pDNA) cellular uptake by 20% in both TNBC cells. Therefore, NK-LNP(PA/pDNA) cellular uptake relied on clathrin-mediated endocytosis by the NKG2D receptor ligand on TNBC. TNBC cell uptake efficiency (%) was determined over time, depending on whether LNP(PA/pDNA) was surface-modified by NK cell membrane protein, by FACS analysis (Appendix A). We confirmed the optimization of uptake efficiency for treatment time and confirmed that it was saturated at 6 h. After 6 h of treatment, NK-LNP(PA/pDNA) showed about 50% cellular uptake efficiency in TNBC cell lines, which showed the greatest difference compared to LNP(PA/pDNA) without NK cell membrane fusion. Next, we determined endosomal escape ability, one of the critical factors in gene delivery, using our LNP delivery system. Figure 2C shows that NK-LNP (RITC-labelled PA/pDNA) was internalized and located near the cell membrane after 1h treatment. After 3 h, endosome (green) and NK-LNP(PA/pDNA) (red) were co-localized (shown in yellow). Afterward, it was confirmed that PA/pDNA escaped from endosomes by showing a separation between green and red signals. Eventually, as shown in Figure 2D, the endosomal escaped PA/pDNA was delivered to the nucleus (Appendix A).

### 3.4. Therapeutic Effect of NK-LNP(PA/pDNA) in TNBC

We determined HIC1 protein expression levels in various breast cancer cell lines (MCF7, SKBR3, BT474, MDA-MB-231, MDA-MB-468, HCC38, and MDA-MB-453) compared to human normal breast cells (MCF12A). It was found that HIC1 expression was suppressed by *HIC1* hyper-methylation in TNBC compared to normal breast cells and other breast cancer cells, which is consistent with previously reported results [12]. Interestingly, *HIC1* hyper-methylation was identified in all TNBC subtype cell lines: basal-like (MDA-MB-468 and HCC38), mesenchymal stem-like (MDA-MB-231), and luminal androgen receptor (MDA-MB-453) cell lines. To confirm the effect of delivered HIC1 pDNA, in vitro experiments were performed first using the C18-ceramide-free LNP system. After treatment, protein levels of HIC1 and downstream pathways were analyzed by Western blot analysis in MDA-MB-468 cells (Figure 3A). Owing to effective targeted delivery, NK-LNP(PA/pDNA) treated cells showed significant overexpression of HIC1 protein compared to PA/pDNA and LNP(PA/pDNA) treated cells (Appendix A). Additionally, to determine whether NK cell membrane proteins affected HIC1-related cell signal pathways, we compared cells treated with NK cell membrane proteins alone and NK-LNPs without PA/pDNA (NK-LNP(bare)). There was no significant change in the expression levels of the HIC1 protein or related downstream pathways in either cell. For cells treated with PA/pDNA, LNP(PA/pDNA), and NK-LNP(PA/pDNA), protein expression levels of SIRT and phospho NF-kB were suppressed due to the HIC1 pDNA delivery. Among these cells, cells treated with NK-LNP(PA/pDNA) showed the highest level of HIC1 expression. This was likely an effect of targeting-based delivery. These results indicated that restored HIC1 and downstream expression patterns were unaltered even when NK-LNP(PA/pDNA) was not incorporated into C18-ceramide (Appendix A). These results revealed that the commercially available lipofectamine delivery system showed slightly higher HIC1 mRNA expression efficiency than the PA/pDNA complex. As expected, NK-LNP using targeting delivery showed an expression efficiency approximately four times higher than that of lipofectamine. We determined the effect of HIC1 restoration by performing cell migration and invasion assay. The migration assay is a gap closure assay that observes cell growth in 2D, while invasion uses the Boyden–Chamber assay to implement contraction and invasion of metastatic cells through the basement membrane within the chamber. Results showed that gap closure after the scratch was significantly reduced in NK-LNP(PA/pDNA)-treated cells compared to untreated cells (Figure 3B). As a control for pDNA, there was little reduction in migration of cells treated with NK-LNP(bare). On the other hand, in cells treated with PA/pDNA and LNP(PA/pDNA), cell migration tended to decrease slightly as HIC1 recovered somewhat. As also expected, NK-LNP(PA/pDNA) treated cells were dramatically reduced (Figure 3C and Appendix A). Invasion ability decreased in inverse proportion to the recovery of HIC1 levels. Amounts of metastasis-related proteins (MMP9) and secreted LCN2 and VEGF proteins as proliferation-related biomarkers were also suppressed (Appendix A). Next, we performed WST-8 analysis to investigate the effects of HIC1 expression and C18-ceramide on TNBC cell viability (Figure 3D). Cell viability decreased to 69% and 45% when treated with C18-ceramide-free LNP(PA/pDNA) and NK-LNP(PA/pDNA), respectively. These results demonstrate that restoring HIC1 can reduce cell migration, invasion, and viability. For NK-LNP(Bare) containing C18-ceramide, cell viability was reduced to 59%. This result means that C18-ceramide-LNPs alone without HIC1 delivery can decrease cell viability. We treated MCF12A (normal breast cells) with NK-LNP(bare), LNP(PA/pDNA), and NK-LNP(PA/pDNA) with C18-ceramide, which are the same samples used for TNBC to confirm cell viability. NK-LNP(bare) is an LNP that is fused only with the NK cell membrane and does not contain a therapeutic agent. It did not show significant cytotoxicity in normal cells, which is believed to be because active targeting does not occur without NK cell membrane fusion (Appendix A). Treatment of cells with NK-LNP(PA/pDNA) containing C18-ceramide dramatically reduced viability by up to 26% in TNBC cells. As a result, it was realized that the treatment effect of TNBC could be synergistically increased when HIC1 was restored and C18-ceramide delivery was used simultaneously. When treated with NK-LNP(PA/pDNA), pro-apoptotic markers (BAX, BAD, and cleaved caspase 3) were increased. In contrast, anti-apoptotic markers (BCL2) and cell proliferation markers (PCNA or Ki67) were decreased (Figure 3E and Appendix A). We also used FACS analysis to cross-confirm the synergistic effect of HIC1 restoration and C18-ceramide. Like cell viability results, NK-LNP(PA/pDNA) with C18-ceramide induced more cell death than without C18-ceramide-LNPs (Figure 3E and Appendix A). Finally, through caspase-3 activity analysis, it was confirmed that TNBC apoptosis was promoted 3.25-fold when combined with HIC1 plasmid and NK-LNP(PA/pDNA) with C18-ceramide (Figure 3F). These results demonstrate that combination therapy of therapeutic chemicals and genes is more effective in treating TNBC than C18-ceramide delivery or HIC1 recovery alone. Molecular biological analysis was conducted to confirm the intracellular therapeutic effect of C18-ceramide. Increased intracellular levels of C18-ceramide upregulated the expression of PP2A (CAPP; ceramide-activated protein phosphatase), which reduced phospho-AKT levels and induced apoptosis in association with TNBC treatment [32,33]. Regulation of NK-LNP and AKT pathways was investigated in the presence or absence of C18-ceramide (Appendix A). C18-ceramide-NK-LNPs without PA/pDNA showed increased expression of PP2A compared to untreated and C18-ceramide-free-NK-LNPs. Downstream signals showed reduced expression levels, including phospho-AKT (S473 and T308) and phospho-S6. Additionally, C18-ceramide-NK-LNPs with PA/pDNA amplified these effects, highlighting their regulatory impact on the AKT pathway. These findings highlight the role of C18-ceramide in influencing PP2A expression and the AKT signaling cascade, revealing potential therapeutic implications for the treatment of TNBC.

### 3.5. Immune Cell Regulation Effect of NK-LNP(PA/pDNA)

To investigate whether NK-LNP(PA/pDNA) could induce antitumor effects in immune cells, human-derived THP-1 (monocyte) cells were differentiated into macrophages (MΦ) and dendritic cells (DC). THP-1 cells were differentiated into M0 macrophages by phorbol 12-myristate 13-acetate (PMA) treatment for 24 h at 37 °C. The morphology of M0 macrophages changed into an adherent cell type, and an increase in CD68 mRNA expression level was measured by qRT-PCR analysis (Appendix A). After NK-LNP(PA/pDNA) treatment, M1 macrophage polarization markers associated with antitumor immunity (iNOS and CD86) were increased, while M2 macrophage markers associated with tumor progression (CD163 and CD206) were decreased compared to M0 macrophages (Figure 4A) [34]. Additionally, ELISA analysis showed that macrophages treated with NK-LNP(PA/pDNA) significantly enhanced M1 macrophage-related cytokines (IL-6 and TNF-α) but decreased M2 macrophage-related cytokines (IL-10) (Figure 4B,C). This suggests that NK-LNP(PA/pDNA) can polarize into M1-macrophages with antitumor effects. Moreover, THP-1 cells (monocytes) were differentiated into immature dendritic cells (iDC) by human recombinant IL-4 and GM-CSF proteins before treatment with NK-LNP(PA/pDNA) for TNBC cells (Appendix A). The differentiation status was confirmed by assessing mRNA levels of DC sign (CD209) via qRT-PCR (Appendix A). Afterward, iDC was co-cultured with TNBC and treated with NK-LNP(PA/pDNA). TNBC cells were induced to release apoptosis-secreted DAMPs (damage-associated molecular patterns). In general, tumor cells induced by apoptosis can release DAMPs in the following order: (i) pre-apoptotic exposure of intracellular calreticulin to the cell membrane surface (ecto-CRT), (ii) early-apoptotic secretion of ATP, and (iii) post-apoptotic release of high-mobility group box 1 (HMGB1) [35]. DAMPs released from co-cultured TNBC can bind to immature DCs in the tumor microenvironment (TME). Additionally, DCs mature within the TME, enhancing antigen presentation capabilities. In mature DCs, levels of inflammatory cytokines such as IL-10, TNF-α, and CD86 in the TME are elevated. These cytokines can maximize antitumor effects by activating surrounding cytotoxic CD8+ T cells [36]. In the results shown in Figure 3, the induction of cell death was more effective with the combination therapy compared to C18-ceramide delivery or HIC1 restoration alone. Similarly, TNBC cells treated with NK-LNP(PA/pDNA) composed of C18-ceramide showed substantial surface-exposed CRT and released ATP and HMGB1 (Figure 4D). DAMPs successfully induced the differentiation of iDCs in mature dendritic cells (mDCs) (Appendix A) [37,38]. Both positive DC sign (CD209) and HLA-DR of mDCs were analyzed by FACS (Figure 4E and Appendix A). In the NK-LNP(PA/pDNA)-treated group, which showed the highest level of DAMP release, the differentiation rate into CD209+ and HLA-DR+ mature dendritic cells was noticeably higher than in the negative control group. Additionally, DCs co-cultured with NK-LNP(PA/pDNA)-treated TNBC cells showed increased secretion of inflammatory cytokines (TNFα, IL-6, and IL-12), affecting antitumor immunity (Figure 4F) [39]. However, measurements of IL-10 secretion did not yield detectable values. These results demonstrated the ability of DCs cultured with NK-LNP(PA/pDNA) to induce antitumor immune responses by observing increased distribution of mDCs and increased secretion levels of TNFα, IL-6, and IL-12. In summary, administering NK-LNP(PA/pDNA) to TNBC patients can induce primary tumor apoptosis. In cascade, the release of DAMPs resulting from the primary death of TNBCs can induce differentiation of nearby immature DCs, promoting their conversion into mature DCs. During this process, macrophages around the tumor can differentiate into M1-macrophages and cause the release of antitumor-related cytokines. Finally, activation of T cells by mature dendritic cells can stimulate immune cell infiltration into the cancer, with immune cells such as CD8 T cells and NK cells within the tumor activated, resulting in a secondary antitumor effect. This comprehensive overview demonstrates the potential of NK-LNP(PA/pDNA) as an immunotherapeutic approach for treating TNBC.

## 4. Discussion

Among breast cancers, the TNBC type has clinical characteristics such as a high metastasis rate, high possibility of recurrence, and poor prognosis. Because TNBC tumors cannot be targeted for endocrine and HER2 therapy, clinically standardized treatment regimens for TNBC are still lacking. Preoperative neoadjuvant chemotherapy, PARP (poly ADP-ribose polymerase) inhibitors, and immunotherapy are being attempted in clinical practice. However, residual metastatic lesions ultimately lead to tumor recurrence, making treatment difficult [40]. Therefore, new TNBC treatment strategies are of clinical urgency. In this study, we determined that the expression levels of HIC1, which are associated with tumor migration, invasion, and metastasis, were associated with TNBC subtypes: basal-like (MDA-MB-468 and HCC38), mesenchymal-like (MDA-MB-231), and luminal androgen receptor (MDA-MB-453). From our results, HIC1 restoration is expected to be a therapeutic strategy for patients with most TNBC subtypes.

Designing specific tumor-targeting carriers and efficiently encapsulating therapeutic agents is a major challenge in developing cancer therapeutics using nanoparticles. In this respect, lipid-based nanoparticles (LNPs) are very attractive carriers. LNPs can encapsulate hydrophilic substances, trap hydrophobic substances between lipids, and modify the LNP surface for receptor targeting. Additionally, LNPs are ideal nanocarriers that can bypass multiple extracellular and intracellular barriers. Essential challenges in the development of therapeutic LNPs include (i) optimizing the state of the encapsulated cargo (drug, gene, or protein), (ii) maintaining solvent stability for cellular uptake, (iii) selectively targeting recipient cells, (iv) effective structuring of the cargo, (v) endosome escape, (vi) cargo transport to the -target organs, and (vii) improvement of treatment environment with immune cells. These issues are addressed by loading anionic plasmid DNA inside the LNPs and modifying the PA protein to increase solvent stability. The modified PA, rich in primary amines, can effectively encapsulate pDNA through NTA-His tag interaction and induce endosomal escape within the cytoplasm. The drug delivery efficiency of LNPs was maximized through the fusion of NK cell membrane protein, which can selectively target TNBC. We demonstrated that combining HIC1 restorations with C18-ceramide delivery can synergistically improve the treatment effect of TNBC. We confirmed that LNPs fused with NK cells can activate immune cells and induce proinflammatory cytokines, which are known to influence antitumor immunity. Our NK-LNP platform could be a potent tumor immunotherapy for difficult-to-treat cancers.

## 5. Conclusions

We demonstrated the therapeutic efficacy of TNBC by simultaneously delivering therapeutic genes and anticancer lipid chemicals via LNP fused with NK cells. Dual therapy combining HIC1 restoration and C18-ceramide uptake may synergistically improve the treatment effect of TNBC. In addition, this was proven to be a combination therapy that could induce anticancer immunotherapy by activating peripheral immune cells due to LNPs’ modification of the NK cell plasma membrane We propose a therapeutic strategy using lipid nanoparticlescould positively promote the development of various cancer treatments. Cell therapy, such as NK cells, is an innovative approach to cancer treatment. This has numerous advantages, including sophisticated targeted treatment and prevention of resistance to anticancer drugs. However, drawbacks such as cost, accessibility, and efficiency uncertainty remain challenges that must be overcome. In this paper, we implemented an LNP platform that mimics NK cells. This will be an alternative that can overcome the limitations of existing cell therapies.

## Data Availability

Data are contained within the article and Appendix A.

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
