# Peer review of "Lipid-Based Nanoparticles Fused with Natural Killer Cell Plasma Membrane Proteins for Triple-Negative Breast Cancer Therapy"

_pharmaceutics, 2024, doi:10.3390/pharmaceutics16091142_

Round 1

Reviewer 1 Report

Comments and Suggestions for Authors

In this manuscript titled “lipid-based nanoparticles mimicking natural killer cells for triple-negative breast cancer therapy” Eun-Jeong Won and colleagues developed a therapeutic liposome nanoparticle loaded with plasmid DNA (to overexpress HIC1) for triple-negative breast cancer. These nanoparticles were also loaded with C18-ceramide (to induce apoptosis) and coated with NK cell membrane proteins. The manuscript needs reversion before publication.

Figures 3 A-C were missing in the manuscript.

These Nanoparticles unfortunately were not tested in animal models of TNBC.

Author Response

Comments 1.

Figures 3 A-C were missing in the manuscript.

: Thanks for the excellent point. This part was an issue that arose during the editing process, and Fig. 3A-C has been added.

Comments 2.

These Nanoparticles unfortunately were not tested in animal models of TNBC.

We fully agree with the reviewer's comments. We performed in vitro experiments using human NK cell lines to confirm the therapeutic effect. We recognize that these results are insufficient to reflect the in vitro immune microenvironment and that additional in vivo experiments are needed to verify this conclusion. For in vivo experiments, a large amount of membrane proteins from mouse NK cells should be isolated and purified to regenerate NK-LNPs. In addition, humanized mice for TNBC xenograft modeling are essential to obtain in vivo data. Currently, we have confirmed the in vitro therapeutic effect using human NK cell lines. At this stage, it takes a long time to confirm the therapeutic effect in an in vivo animal model.

 We have changed the expression "regulates the tumor immune microenvironment" in the main text to "inhibits tumor growth."

 We agree with the reviewer's comments and will study the process of isolating and purifying sufficient mouse NK cell membrane proteins and creating a TNBC xenograft model in subsequent studies. In addition, we will administer NK-LNP (PA/pDNA) to mice according to the treatment proposed in this paper to analyze the tumor suppression effect and immune cell response. We sincerely appreciate your valuable advice.

Reviewer 2 Report

Comments and Suggestions for Authors

This manuscript prepared a lipid-based nanoparticle, which had the similar structure to NK cells. The lipid-based nanoparticle can induce tumor apoptosis and invade metastasis. However, there are some problems in the description of the manuscript and the designation of experiments:

1. The manuscript described LNPs as mimic NK cells, because LNPs were loaded with plasmid DNA, decorated with NK membrane protein, and had kill activity to some degree. However, these characterizations can’t be worthy of mimicking NK cells, which should be acquired with immune functions rather than only structure. Therefore, please replace statement with other appropriate one.

2. Please write the designation of LNPs clearly in the abstract, including HIC1 DNA, C18-ceramide, and NK cell membrane coating.

3. In F2 CD images, more cells in the same field of view are required to demonstrate lysosomal escape and nuclear localization of LNP, and a small number of cells in different fields are not convincing. In addition, please supplement separate images for each channel.

4. There are no accurate data in the manuscript quantifying the proportion of cells transfected with HIC1 DNA, and this conclusion cannot be drawn solely from the nuclear localization images. Transfection efficiency is an important factor that needs to be taken into account in the killing tests.

5. Please check the WB data of the full manuscript:

1) F1F lacks actin bond. Please explain the reason for choosing 1:5 (lipid: protein).

2) Please quantify the data, like F3E WB date.

3) For data validating H1C expression and mechanistic pathway, please supplement multiple batches of experiments (n≥3), and conduct significant difference analysis. Because the existing relevant data do not observe a significant difference.

4) Please explain the reason for incubation time, whether 6 hours ensure DNA be uptaken and expressed.

6. F3 A-B data were not visible.

7. In the killing assay in vitro, the data for F3D and F3F do not correspond. More importantly, LNPs do not seem to exhibit outstanding ability to induce tumor cells apoptosis, so it may not be able to support the anti-tumor view. In this regard, increasing the co-incubation time may affect the data results.

8. For the safety of LNPs, please supplement the experiment to support the view that the specific killing and transfection ability of LNPs.

9. In the part of “3.5. Immune cell regulation effect of NK-LNP(PA/pDNA)”, the research on the regulatory function of immune cells needs to be verified by experiments in vivo, and the co-incubation experiments in vitro cannot reflect the real immune microenvironment. Therefore, please supplement the experiment in vivo to verify this conclusion. Moreover, T cells are important response cells for immune activation and should also be included in immunity analysis.

10. There are some inaccuracy in the manuscript:

1) Please unify the nomenclature of LNP in the full text, NK-LNP or NKM-LNP.

2) Please replace the subtitle of Figure2, which could be attach to conclusion in this part.

3) Please supplement the scale of F1B.

4) The correspondence between NK-LNP and Membrane can not be clearly observed in F1E, please replace the picture with a more representative one.

5) In the Introduction part, “We loaded plasmid DNA encoding HIC1 inside the LNPs, which can effectively introduce NK cell membrane proteins onto the particle surface and inhibit cancer metastasis”, please correct the statement. 

6) In the F4D, CAT isn’t the secret protein, please correct it. 

Comments on the Quality of English Language

English need to be improved. 

Author Response

Comments 1. The manuscript described LNPs as mimic NK cells, because LNPs were loaded with plasmid DNA, decorated with NK membrane protein, and had kill activity to some degree. However, these characterizations can’t be worthy of mimicking NK cells, which should be acquired with immune functions rather than only structure. Therefore, please replace statement with other appropriate one.

: Thank you for pointing out the correct wording. We changed the expression mimicking NK cells to 'Lipid-based nanoparticle fused with NK cell plasma membrane proteins' in title and main text.

Comments 2. Please write the designation of LNPs clearly in the abstract, including HIC1 DNA, C18-ceramide, and NK cell membrane coating.

: Thanks for pointing out that the abstract could have been written more clearly. In the abstract, I wrote more clearly about each component included in the LNP (highlighted in blue text) as below.

“LNPs fused with NK cell membrane protein system (NK-LNP) have three main features: i) hydrophilic plasmid DNA can inhibit TNBC metastasis when encapsulated within LNPs and delivered to cells; ii) the lipid composition of LNPs, including C18 ceramide, exhibits anticancer effects; iii) NK cell membrane proteins are immobilized on the LNP surface, enabling targeted delivery to TNBC cells. These particles facilitate the targeted delivery of HIC1 plasmid DNA and the modulation of immune cell functions.”

Comments 3. In F2 CD images, more cells in the same field of view are required to demonstrate lysosomal escape and nuclear localization of LNP, and a small number of cells in different fields are not convincing. In addition, please supplement separate images for each channel.

: Thank you for pointing out that we can provide data. We added low-magnification images and each channel image to Figure S5C and D as below.

Comments 4. There are no accurate data in the manuscript quantifying the proportion of cells transfected with HIC1 DNA, and this conclusion cannot be drawn solely from the nuclear localization images. Transfection efficiency is an important factor that needs to be taken into account in the killing tests.

: Thank you for your precise point. We have confirmed the transfection quantifying. Basically, we used a form of HIC1 with GFP linked to the C-terminus to distinguish it from the natively expressed HIC1 protein in TNBC. This is included in Figure S6, and the relevant analysis conditions have been added.

Comments 5. Please check the WB data of the full manuscript:

1) Fig1F lacks actin bond. Please explain the reason for choosing 1:5 (lipid: protein).

: We apologize for the confusion during the review. We have added a detailed explanation to the Figure 1F caption.

We isolated and purified membrane proteins from NK cell lines and fused them to LNPs. Therefore, if the isolation and purification are successful, actin, which is usually present in the NK cytoplasm, will not be expressed in WB. In addition, we performed fusions at quantitative ratios of 1:5 and 1:10, and the amount fused to LNPs was similar. Therefore, the material manufactured in the subsequent process was used at a ratio of 1:5.

2) Please quantify the data, like Fig3E WB date.

: Thanks for pointing out the exact data needed.

We quantified the western blot data and added it to Figure S7D.

3) For data validating H1C expression and mechanistic pathway, please supplement multiple batches of experiments (n≥3), and conduct significant difference analysis. Because the existing relevant data do not observe a significant difference.

: Thank you for your feedback on data reproducibility. Basically, we do three WB analyses. Below are the data.

The WB data is repetitive, so we did not add it to the main text or supplementary materials. However, if the reviewers deem it necessary, we will add the following replicate data to the supplementary materials.

4) Please explain the reason for incubation time, whether 6 hours ensure DNA be uptaken and expressed.

: We added a description of the incubation time to the main text.

“We confirmed the efficiency of uptake and expression over time. The efficiency was saturated when incubated for 6 hours, which was the optimal treatment condition.”

Comments 6. Fig3 A-B data were not visible.

: Thanks for the excellent point. This part was an issue that arose during the editing process, and Fig. 3A-C has been added.

Comments 7. In the killing assay in vitro, the data for Fig3D and Fig3F do not correspond. More importantly, LNPs do not seem to exhibit outstanding ability to induce tumor cells apoptosis, so it may not be able to support the anti-tumor view. In this regard, increasing the co-incubation time may affect the data results.

: Thank you for pointing out that the two data do not match. Our explanation was insufficient.

Fig3D and 3F are samples treated at equal times in the experimental group. The only difference is the time point of analysis. Fig3D is the result of analysis at 72 hours after treatment, and Fig3F is the result of analysis at 48 hours after treatment. The reason for analyzing 48 hours in Fig3F is that when Annexin-V/Propidium iodide staining was performed and analyzed 72 hours later, cell death was so severe that FACS analysis was difficult. In conclusion, the mortality rates of the experimental groups 3D and 3F were similar patterns because the time point of analysis after treatment differed, but there was a difference in the quantitative aspect.

Comments 8. For the safety of LNPs, please supplement the experiment to support the view that the specific killing and transfection ability of LNPs.

: We conducted a toxicity evaluation of NK-LNP itself. This is described in Figure S9. However, the description of NK-LNP(bare) was insufficient. Thank you for your advice.

We have expressed it more clearly in the main text.

"We treated MCF12A(normal breast cell) with NK-LNP(bare), LNP(PA/pDNA), and NK-LNP(PA/pDNA) with C18-ceramide, which are the same samples used for TNBC to confirm cell viability. NK-LNP(bare) is an LNP that is fused only with the NK cell membrane and does not contain a therapeutic agent. It did not show significant cytotoxicity in normal cells, which is believed to be because active targeting does not occur without NK cell membrane fusion (Figure S9)."

Comments 9. In the part of “3.5. Immune cell regulation effect of NK-LNP(PA/pDNA)”, the research on the regulatory function of immune cells needs to be verified by experiments in vivo, and the co-incubation experiments in vitro cannot reflect the real immune microenvironment. Therefore, please supplement the experiment in vivo to verify this conclusion. Moreover, T cells are important response cells for immune activation and should also be included in immunity analysis.

: We fully agree with the reviewer's comments. We performed in vitro experiments using human NK cell lines to confirm the therapeutic effect. We recognize that these results are insufficient to reflect the in vitro immune microenvironment and that additional in vivo experiments are needed to verify this conclusion. For in vivo experiments, a large amount of membrane proteins from mouse NK cells should be isolated and purified to regenerate NK-LNPs. In addition, humanized mice for TNBC xenograft modeling are essential to obtain in vivo data. Currently, we have confirmed the in vitro therapeutic effect using human NK cell lines. At this stage, confirming the therapeutic effect in an in vivo animal model takes a long time.

We have changed the expression "regulates the tumor immune microenvironment" in the main text to "inhibits tumor growth."

We agree with the reviewer's comments and will study the process of isolating and purifying sufficient mouse NK cell membrane proteins and creating a TNBC xenograft model in subsequent studies. In addition, we will administer NK-LNP (PA/pDNA) to mice according to the treatment proposed in this paper to analyze the tumor suppression effect and immune cell response. We sincerely appreciate your valuable advice.

Comments 10. There are some inaccuracy in the manuscript:

1) Please unify the nomenclature of LNP in the full text, NK-LNP or NKM-LNP.

: We are infinitely grateful for the precise expressions.

All expressions in the text have been corrected to ‘NK-LNP’

2) Please replace the subtitle of Figure2, which could be attach to conclusion in this part.

: Thank you for helping us improve the expression to be more precise. We have modified the subtitle.

"Figure 2. Specific targeting and cellular uptake mechanism of NK-LNP(PA/pDNA) in cells."

3) Please supplement the scale of F1B.

: Thanks for pointing out that We could provide more accurate information.

We added a scale bar to Figure 1B.

4) The correspondence between NK-LNP and Membrane can not be clearly observed in F1E, please replace the picture with a more representative one.

: We agree with what you pointed out. We repeated this data several times, and we could only get smear band results each time like the data presented in Figure 1E. This is because we wanted to observe many proteins besides the NK cell membrane proteins. In this regard, the references (Kang M. et al., Adv Mater. 2020. / Pitchaimani A. et al., Biomaterials. 2018. / Deng G. et al., ACS Nano. 2018) that reported the SDS-PAGE results after purifying NK cell membrane proteins also showed similar results to ours. Unfortunately, we cannot obtain clear bands for samples containing many proteins with our current technology.

5) In the Introduction part, “We loaded plasmid DNA encoding HIC1 inside the LNPs, which can effectively introduce NK cell membrane proteins onto the particle surface and inhibit cancer metastasis”, please correct the statement. 

: Thank you for pointing out the correct expression.

We have corrected it in last paragraph of Introduction part.

6) In the Fig4D, CAT isn’t the secret protein, please correct it. 

: Thank you for your accurate comment.

CRT is not a secreted protein, but a plasma membrane exposure protein when cell death is induced. We corrected it in the main text.

Reviewer 3 Report

Comments and Suggestions for Authors

In my opinion, the paper would benefit from additional information and experiments.

In my opinion, Authors should refrain from calling their particles LNPs as they are strikingly different from the approved mRNA and siRNA LNPs. The particles described in the paper are closer to lipopolyplexes (Huang Leaf's lab).

Authors need to quantify the the amount of NK-derived proteins on LNPs (ng protein / mg lipid) and, demonstrate batch to batch reproductibility.

DNA encapsulation should be quantitated using SYBR green as the lower sensitivity of gel electrophoresis will overestimate encapsulation efficiency.

To validate targeting, Authors need to add a group with LNps coated with plasma proteins from irrelevant cells (eg reticulocytes).

Authors need to detail the imaging method. Fig. 2D should be completed with statistical analysis.

As cancer associated fibroblasts are a dominant cell population in TNBC, Authors could include them for targeting evaluation.

Data in Fig. S6 do not show HIC1 hypermethylation.

Authors need to introduce ceramide, polyarginine, NTA-Ni.

Comments on the Quality of English Language

some mistyakes in english

Author Response

Comments 1.

In my opinion, Authors should refrain from calling their particles LNPs as they are strikingly different from the approved mRNA and siRNA LNPs. The particles described in the paper are closer to lipopolyplexes (Huang Leaf's lab).

: We agree with your opinion. In general, LNP (Lipid Nanoparticle) is composed of multiple layers of lipids and can encapsulate active ingredients such as drugs, mRNA, and siRNA inside. LNP is a term applied to materials that are composed of solid lipid particles and liquid lipid layers and can effectively protect and deliver drugs.

We have a different structure from general LNP. Therefore, we agree that other expressions, such as liposome, are more appropriate than LNP. Nevertheless, we expressed LNP as an abbreviation for ‘lipid-based nanoparticles’ in the main text.

Please let me know if you need to clarify the expression more. We will revise it again.

Comments 2.

Authors need to quantify the amount of NK-derived proteins on LNPs (ng protein / mg lipid) and, demonstrate batch to batch reproductibility.

: Thank you for your accurate comment.

We confirmed that it was saturated at the final ratio of 1:5. The quantitative value at this time is included in the description of Figure 1E in the main text. The quantitative value was entered by repeating the experiment at least 3 times.

Comments 3.

DNA encapsulation should be quantitated using SYBR green as the lower sensitivity of gel electrophoresis will overestimate encapsulation efficiency.

: Thank you for your comment on the quantification method.

We have already confirmed DNA encapsulation with SYBR green. To quantify the loaded DNA, the LNPs must be treated with triton-x100 after manufacturing to destroy the structure. In this process, it was difficult to obtain reproducible results when quantifying with SYBR green. This was thought to be due to interference between the surfactant and the assay tool used. Therefore, as a supplementary method, we confirmed it with electrophoresis, which has little interference. Please tell us again if the reviewer determines that quantification with SYBR green is necessary. We will try to find conditions without interference with the surfactant.

Comments 4.

To validate targeting, Authors need to add a group with LNps coated with plasma proteins from irrelevant cells (eg reticulocytes).

: Thank you for your advice on targeting verification.

We confirmed that targeting efficiency was very low when NK-LNP was treated with the same conditions as the negative cell line MCF12A (normal breast cells) for targeting verification. In addition, we confirmed which protein among the NK cell membrane proteins on the LNP surface was involved in cell targeting. The targeting efficiency was significantly reduced when the NK cell membrane protein on the LNP surface was blocked with a specific NKG2D ligand and treated with the positive cell line (MDA-MB-231, MDA-MB-468). The comprehensive results confirmed that NKG2D components on the LNP surface were involved in targeting.

We did not perform this because NKG2D is a plasma protein of NK cells and is not present in unrelated cells such as reticulocytes. It is a suitable verification method to perform targeting verification using membrane proteins obtained from cells other than NK cell lines, but we do not think it is necessary because we concluded that targeting is due to NKG2D.

Comments 5.

The authors need to detail the imaging method. Fig. 2D should be completed with statistical analysis.

: Thank you for your suggestions for improving the paper.

As you pointed out, we added 'Fluorescence image analysis' to the main text method section and explained it. We also added statistical analysis data to Fig. 2D.

Comments 6.

As cancer associated fibroblasts are a dominant cell population in TNBC, Authors could include them for targeting evaluation.

: Thank you for your helpful comments on the targeting evaluation.

We were not able to perform all the dominant cell populations of TNBC. However, we performed the experiment based on previous studies.

TNBC has various subtypes such as BL1, BL2, IM, M, MSL, LAR, and unstable (Li, C.J.; Tzeng, Y.D.T.; Chiu, Y.H.; Lin, H.Y.; Hou, M.F.; Chy, P.Y. Pathogenesis and Potential Therapeutic Targets for Triple-Negative Breast Cancer. Cancers (Basel). 2021, 13, 2978). Many researchers have pointed out that no specific subtype is dominant and exists in similar populations. Therefore, we selected the known non-overlapping subtypes MDA-MB-231 and MDA-MB-468 for the experiment. However, we think the reviewer's recommendation is a meaningful point. Therefore, we will evaluate the targeting efficiency of cancer-associated fibroblasts in future experiments.

Comments 7.

Data in Fig. S6 do not show HIC1 hypermethylation.

: Thank you for your detailed comments.

The data in Fig S6 are the results of confirming HIC1 hypermethylation expressed in TNBC before LNP treatment. As the reviewer pointed out, HIC1 hypermethylation is not confirmed in wild-type TNBCs compared to normal breast or other breast cancers.

Comments 8.

Authors need to introduce ceramide, polyarginine, NTA-Ni.

: Thank you for pointing this out and improving the quality of the paper. I have added a description of the respective roles of ceramide polyarginine, and NTA-Ni in the introduction and results section.

Round 2

Reviewer 1 Report

Comments and Suggestions for Authors

Since the authors have addressed the comments in the revision, the manuscript now can be published.

Reviewer 2 Report

Comments and Suggestions for Authors

The author has solved my problem. 

Comments on the Quality of English Language

Good. 

Reviewer 3 Report

Comments and Suggestions for Authors

Authors adressed most comments, the comments not adressed require additional experiments that, in my opinion, are related to validation of targeting. I do recommend acceptance of the revised paper.